# Zinc Deficiency and Zinc Supplementation in Allergic Diseases

**DOI:** 10.3390/biom14070863

**Published:** 2024-07-19

**Authors:** Martina Maywald, Lothar Rink

**Affiliations:** Institute of Immunology, Faculty of Medicine, RWTH Aachen University Hospital, 52074 Aachen, Germany; martina.maywald@rwth-aachen.de

**Keywords:** zinc biology, nutrition, dietary fiber, allergy, eosinophilia, asthma, chronic rhinosinusitis, atopic dermatitis, food allergy

## Abstract

In recent decades, it has become clear that allergic diseases are on the rise in both Western and developing countries. The exact reason for the increase in prevalence has not been conclusively clarified yet. Multidimensional approaches are suspected in which diet and nutrition seem to play a particularly important role. Allergic diseases are characterized by a hyper-reactive immune system to usually harmless allergens, leading to chronic inflammatory diseases comprising respiratory diseases like asthma and allergic rhinitis (AR), allergic skin diseases like atopic dermatitis (AD), and food allergies. There is evidence that diet can have a positive or negative influence on both the development and severity of allergic diseases. In particular, the intake of the essential trace element zinc plays a very important role in modulating the immune response, which was first demonstrated around 60 years ago. The most prevalent type I allergies are mainly based on altered immunoglobulin (Ig)E and T helper (Th)2 cytokine production, leading to type 2 inflammation. This immune status can also be observed during zinc deficiency and can be positively influenced by zinc supplementation. The underlying immunological mechanisms are very complex and multidimensional. Since zinc supplements vary in dose and bioavailability, and clinical trials often differ in design and structure, different results can be observed. Therefore, different results are not surprising. However, the current literature suggests a link between zinc deficiency and the development of allergies, and shows positive effects of zinc supplementation on modulating the immune system and reducing allergic symptoms, which are discussed in more detail in this review.

## 1. Introduction

Allergic diseases are a series of disorders characterized by an excessive immune response to substances that are harmless to the human body at first glance. This so-called hypersensitivity of the immune system is directed against a biological or chemical substance that triggers an allergic immune response, leading to allergic respiratory diseases such as asthma and allergic rhinitis (AR) or allergic skin diseases such as atopic dermatitis (AD).

In Westernized countries, asthma and atopic disease are public health concerns because of their high prevalence of 10–25%, associated morbidity, and substantial health care and societal costs [1,2,3]. There is an alarming upward trend in the prevalence of allergies in developing countries, possibly due to a shift in lifestyle towards Western habits [4,5].

In Europe, for example, more than 128 million people are affected by allergic diseases, with up to 30% of younger Europeans [2]. In America, almost a third of adults aged 18 and over have a seasonal allergy, food allergy, or eczema [6]. In the case of atopic dermatitis, up to 30% of preschool-age children, 15–20% of school-age children, and 7% of adults are affected, with an economic impact comparable to that of asthma [1]. Nearly 25 million people, or about 13% of the US population, suffer from asthma. The direct and indirect costs to the US economy are estimated at approximately 56 billion dollars annually [7]. The World Health Organization (WHO) declared asthma to be one of the most important non-communicable diseases worldwide. In 2019, an estimated 262 million people worldwide suffered from asthma, resulting in 455,000 deaths [8].

The alarming rise of the prevalence of asthma and atopic disease since the 1970s in Western countries like Europe, Australia, and North America [1,3] can also be observed in the Asian population, like in Japan [3,9,10]. Moreover, developing countries are increasingly burdened by the occurrence of allergies during the last decades [4,5], which makes the need for research into the causes very urgent. Despite many studies into the causes of allergies, their development remains partially misunderstood and many different mechanisms and factors seem to be involved, like changed environmental exposure due to climate change and air pollution by pollen, ozone, nitrogen oxides, and ultrafine particles [2], and Westernization of dietary patterns [1]. Since the current literature suggests a link between nutrition and development or alleviation of allergies/allergy symptoms, but the individual studies provide very different, sometimes contradictory results, which can be confusing, this article was written to shed light on the subject.

To distinguish allergic hyper-responsive immune responses, the classification by Cooms and Gell can be used, dividing these into four subtypes according to the type and effector mechanism responsible for cell and tissue injury: Type I—immediate or immunoglobulin (Ig)E mediated; Type II—cytotoxic or IgG/IgM-mediated; Type III—IgG/IgM immune complex-mediated; and Type IV—delayed-type hypersensitivity or T cell-mediated [11]. Most allergic diseases recognized in the population are caused by aberrant IgE production, involving activation of effector cells, mainly mast cells and basophils/eosinophils, which lead to inflammatory responses and clinical symptoms such as red, itchy eyes, sneezing, nasal congestion, runny nose, coughing, and itchy, swollen skin [11]. It is currently not yet fully understood why some people develop allergies and others do not.

Moreover, the relationship between chronic airway hyper-responsiveness and airway inflammation seems to be not fully elucidated yet. Studies uncovered that there seems to be only a weak correlation at baseline between eosinophilic inflammation and bronchial hyper-responsiveness [12,13]. The anti-IgE monoclonal omalizumab reduces airway eosinophilia but has no effect on bronchial hyper-responsiveness [14], whereas the anti-TNF monoclonal etanercept reduces hyper-responsiveness but has no effect on airway inflammation [15]. Furthermore, there is no relationship between the extent of airway remodeling, specifically reticular basement membrane thickness, and the degree or duration of any inflammatory parameter [16]. Hence, regarding the role of eosinophilic inflammation and the severity of the disease, further research appears to be required to uncover the involved mechanisms.

In general, it is assumed that a combination of environmental risk factors, hygiene standards, and a genetic predisposition to develop allergies (so-called atopy) are responsible for development of allergic diseases [2,17,18]. A genetic influence has been suspected for decades, as epidemiological findings indicate that if one parent suffers from allergies, 33.33% of their children will develop an allergy, and if both parents are affected, this number is 60–70% [17]. It is also known that atopic people produce significantly higher amounts of IgE compared to nonallergic individuals, leading to abnormal immune responses, especially at barrier sites in the body, like the gastrointestinal tract or respiratory tract, which are crucial to protect the body’s internal milieu from the external environment [19].

One crucial process in developing allergies might be immunological imprinting, which refers to the preference of the immune system to recall existing memory cells and not to stimulate de novo reactions in the event of contact with a new but related antigen [20]. The priming of the immune system is particularly effective during the early period of life [21]. Hence, life stages such as the prenatal, perinatal, and early postnatal periods are crucial for the development of a balanced immune system. During these stages, the establishment of a balanced gut microbiota has a key role in reducing allergy development [22,23]. The importance of immunological imprinting is highlighted by clinical studies, where early probiotic applications given to mothers prevented allergies in the offspring [24,25,26]. Hence, a lack of bacteria in early childhood is shown to lead to immunological defects that can persist into adulthood and increase susceptibility to chronic inflammatory diseases like allergies [27,28].

In line with this, atopic diseases are generally associated with the first decades of life and thus with the maturation of the immune system [29]. Nevertheless, late and individual courses of disease are also observed. The first specific IgE antibody reactions are observed just a few months after birth and mainly affect food proteins, especially chicken eggs and cow’s milk [30]. In contrast, sensitization to indoor and outdoor environmental allergens, such as dust mites, pet hair, fungi, and pollen, requires more time and can normally be observed between the first and tenth year of life. The level of exposure is a critical factor and is decisive for the annual incidence of sensitization. About 30 years ago, a German study already demonstrated a dose–response relationship between early exposure to house-dust mites and cat allergens and the risk of sensitization in the first years of life [31]. Moreover, strong IgE antibody responses of infants to food proteins are shown to be markers of atopic reactivity that could predict later sensitization to aeroallergens [32,33].

Since there are hardly any possibilities for an etiologically oriented treatment of allergies, prevention is of particular importance [34]. In this regard, it is essential to be aware of possible risk factors like nutrition. The Western diet is considered an environmental risk factor for developing allergic diseases, while the Mediterranean diet has been shown to be protective [4,5,35]. Dietary factors not only affect the development of allergic diseases, but also influence disease course and severity. High intakes of energy, saturated fat, and protein, and a low fiber intake, increase the risks of asthma and allergic rhinitis (AR) [36]. In contrast, high consumption of vegetables and fruits, olive oil, and fish lowers risks and exacerbates the severity of asthma and AR [36,37,38]. Moreover, adequate intake of vitamins, minerals, and trace elements like zinc is associated with a lower risk of the development of atopic diseases and a reduction in symptoms [39].

## 2. Role of Zinc in Allergic Disease Development and Exacerbation

Nutrition plays an important role in supporting a robust immune system. Malnutrition impairs immune function, leading to a variety of disorders or impairments of the immune system [40]. An increasing number of people are currently susceptible to mineral deficiencies, including older individuals, who often suffer from chronic diseases, and people strictly adhering to restrictive or unbalanced diets often seen in vegetarians, vegans, pregnant women, or athletes forcing weight loss [41].

Since the 1960s, zinc has been known to be an important modulator for the innate and adaptive immune system and to maintain immune tolerance. Symptoms of zinc deficiency are variable and often unspecific, like mental disturbances, frequent infections, depressed immune function, and growth retardation [42]. Zinc deficiency can be attributed to several causes, including low intake due to certain diets or general malnutrition; high consumption of non-zinc dietary components like phytate, which prevents zinc absorption via the intestine; or simultaneous supplementation of high amounts of iron, which negatively influences zinc absorption [43,44], especially if administered in liquid form [45]. Furthermore, chronic drug usage can lead to zinc malabsorption via the intestine, as seen with, amongst others factors, proton pump inhibitors (PPIs), penicillamine, various diuretics, and certain antibiotics [46].

Until now, there have been conflicting results regarding the development of allergic diseases and prenatal prophylaxis by maternal zinc supplementation during pregnancy. Some studies reveal a beneficial correlation of maternal dietary zinc intake during pregnancy and allergic disease development in the offspring [47,48,49], as seen for asthma, wheezing, and atopic dermatitis during childhood, whereas others found no direct correlation [50,51,52,53]. Nevertheless, zinc deficiency is considered a risk factor for allergic diseases being involved in development and exacerbation, as shown by multiple clinical studies [54,55,56]. On the molecular level, the effects of zinc deficiency are highly complex since it affects multiple immune cells and alters a multitude of immunological pathways and protein expression patterns that eventually cause aberrant and disturbed immune function.

T cell development and maturation is highly dependent on zinc, and zinc deficiency is known to cause thymic atrophy and reduced activity of the serum hormone thymulin, which is necessary for T cell maturation and differentiation. In particular, the T helper (Th)1/Th2 ratio is imbalanced by zinc deficiency, showing a reduction in Th1 cells, which are responsible for elimination of intracellular pathogens and supporting cellular immunity. Th2 cells are outnumbered, leading to altered humoral immunity by activating B cells and antibody production, which are mainly involved in allergic diseases [57]. Additionally, zinc deficiency is known to increase allergic eosinophilic inflammation, as present for example in allergic asthma [58,59], by favoring the Th2 cell response and Group 2 innate lymphoid cells (ILC2s); these are critical upstream mediators of type 2 inflammation, which induced airway eosinophilia in translational models of allergic asthma [60,61,62]. Although ILC2s seem to be important in allergic eosinophil immune responses, little is known about the mutual interactions between eosinophils and ILC2s despite their co-occurrence in allergic diseases [63].

Moreover, dendritic cells (DCs), which capture allergens that enter the epithelial barriers of the human body, play an important role in Th2 differentiation. Zinc homeostasis is important for DC differentiation, since zinc-activated proteins, like A20, are necessary to suppress gene expression [64]. In case of zinc deficiency, DC maturation and antigen presentation is enhanced, whereas the tolerogenic phenotype is suppressed, leading to a disturbed adaptive immune response against pathogens [65,66]. Hence, secretion of Th2-cell related cytokines like interleukin (IL)-4 can be reinforced, supporting the pathogenesis of allergic diseases. This reaction is intensified by epithelial cells, which also produce a cytokine milieu that promotes the generation of Th2 cells due to epithelial barrier damage [67]. This results in high levels of IL-4 and IL-13 triggering immunoglobulin (Ig)E isotype class-switching in B cells. Consequently, IgE binds through high-affinity FcεRI receptors on the surface of specific innate effector cells, like mast cells. During an acute encounter with an allergen, cross-linking of the IgE on the surface of mast cells is induced, which triggers the release of mediators including preformed histamine to generate acute symptoms such as itching, sneezing, coughing, and diarrhea in mucosal tissues [68]. There is evidence that differentiation of mast cells within the bone marrow and other tissues is elevated during zinc deficiency and these mast cells contain increased numbers of specific granules; however, recent data are very limited [66,69]. Additionally, IgE secretion is elevated during zinc deficiency [55,70]. Nevertheless, the complete mechanism does not seem to be fully elucidated yet since contradictory studies found zinc chelation to inhibit the histamine release of mast cells [71].

## 3. Zinc and Respiratory Allergic Diseases

Chronic allergic respiratory diseases, such as asthma or rhinosinusitis (CRS), are among the most common health problems in modern countries, being responsible for many visits to the doctor’s office. This results in a reduced quality of life for the people affected by CRS or asthma [1,72]. The exact etiology is not clear yet, which complicates the treatment of patients with chronic allergic airway disease, as different classes of drugs are used. These include anti-inflammatory drugs and systemic or topical corticosteroids, which do not address the cause of the diseases [72]. Several risk factors for respiratory allergic diseases have been known for a long time, e.g., a family history, air pollution, working with animals, dust, and obesity. More recently, it has also been found that micronutrient imbalances can lead to impaired immune defense mechanisms against oxidative stress and pro-inflammatory stimuli [73,74,75]. Malnutrition in middle- and low-income countries, and especially in developing countries, is a heavy social burden that makes allergies more likely to occur [4,5,76].

Furthermore, growing up in a more hygienic environment is believed to increase the risk of developing allergies [77], as affirmed by epidemiological studies showing that overcrowding and unhygienic conditions were associated with a lower prevalence of allergies, eczema, and asthma [78]. Studies assume that increased exposure to microbes early in life may prevent allergic diseases, and therefore a reduced exposure to microbes may explain the rise in allergic diseases especially seen in industrialized countries [18,79]. Parasitic organisms such as helminths are an important aspect of the hygiene hypothesis as they cause chronic infection, suggesting successful downregulation of the immune system, leading to reduced immunopathology and a lower severity of allergic diseases [80,81]. In line with this, helminths are known to suppress Th2 cell responses and lower IgE blood levels and the quantity of eosinophilia [81,82]. It is therefore reasonable to assume that this may be one of the reasons why the prevalence of allergies is lower in developing countries.

Another important aspect in the development of allergies is zinc deficiency. It has long been known that zinc deficiency weakens and alters the immune defense and leads to, among other things, an increased Th2 cell response, elevated eosinophilia [57,58,59], and the release of pro-inflammatory cytokines, e.g., IL-4, IL-6, leukotriene B4 (LTB4), and prostaglandin E2 [57,83]. In addition, zinc-mediated antioxidant activity in the lungs is disrupted, which may be responsible for the imbalance between oxidation processes and antioxidant activity and increases the risk of asthma [84]. The airway epithelium is more susceptible to apoptosis during zinc deficiency, which might be due to elevated caspase activity [85,86], which eventually leads to weakened epithelium barrier function [87] and worsened airway inflammation and airway hypersensitivity [76,88]. Zinc supplementation shows many positive effects on impaired immune function during zinc deficiency: the Th1/Th2 cell balance can be restored [57,83]; mast cell degranulation can be inhibited, consequently diminishing histamine production [89]; apoptosis of epithelial cells is reduced [90]; the overall antioxidant deference of the immune system is strengthened [91]; inflammation is reduced [83]; caspase activity is inhibited [92]; and eosinophil influx into the airways and circulating eosinophils are reduced [93]. Moreover, the pulmonal and gastrointestinal barrier structure and function is improved by zinc supplementation by modulation of intercellular junctional complexes (see Figure 1), which will be explained in detail later. Consequently, allergic symptoms can be weakened.

In clinical studies, adequate zinc levels during pregnancy predict better lung function in the offspring and a lower risk of developing asthma in childhood [47,48,94,95]. In contrast, zinc deficiency increases the risk of developing bronchial reactivity and allergy-like symptoms by up to five times [75,96]. Accordingly, reduced serum or sputum zinc concentrations have been found in asthmatic adults [55,89,97,98,99,100] and children [93,95,101,102,103,104] (see Table 1), who also exhibit increased oxidative stress and airway inflammation [103,105]. In addition, studies have shown that zinc deficiency is associated with severe asthma, increased asthma attacks [96], and decreased lung function, as determined by respiratory parameters such as forced expiratory volume in one second (FEV1) and the ratio of forced expiratory volume in one second to forced vital capacity (FEV1/FVC). The marked reduction in these parameters during zinc deficiency indicates poor lung function in the subjects [105].

A study by Richter et al. uncovered that zinc deficiency significantly increases the level of bronchopulmonary eosinophilia by 35%. Consistent with this, an increase in eosinophils was found in the perivascular and peri-bronchial regions of the lung. In contrast, zinc supplementation was found to decrease BAL eosinophils by 34% [58]. In line with this, Ehlayel et al. showed that hypozincemia, which was observed in 42 (25%) children, was associated with elevated IgE levels [106]. However, as mentioned in Section 1, studies uncovered that there seems to be only a weak correlation between eosinophilic inflammation and bronchial hyper-responsiveness, and more research will help to uncover the underlying molecular mechanisms [12,13,16].

Zinc supplementation, on the other hand, can alleviate symptoms such as wheezing, cough, and dyspnea, and improve lung parameters such as FVC, FEV1, and the FEV1/FVC-ratio [90]. In line with this, studies showed a protective role of zinc in asthma prevention and symptom improvement [94,95,107,108,109,110,111,112,113,114] (see Table 2). Rerksuppaphol et al. found that the pediatric respiratory assessment measure (PRAM) significantly improves following zinc supplementation in the first 48 h; however, there was no difference at the end of the study between the two groups [108]. Ghaffari et al. revealed an improved clinical manifestation and pulmonary function test in asthmatic patients due to zinc supplementation [109].

In contrast, some studies have found no effect of zinc deficiency on allergic asthma [74,93,115,116,117,118,119,120,121,122], or increased values compared to healthy controls [123], indicating that the relationship between zinc and asthma may not yet have been conclusively established and that study results are highly dependent on clinical trial design.

### Zinc and Chronic Rhinosinusitis

Another allergic respiratory disease is Chronic Rhinosinusitis (CRS). CRS is clinically classified into two subtypes depending on the presence (CRSwNP) or absence of nasal polyps (CRSsNP) [124]. However, studies examining CRS do not necessarily distinguish between CRS with nasal polyps (CRSwNP) or without nasal polyps (CRSsNP) [125]. Although the pathogenesis of CRS has not been fully elucidated yet, patients with CRS are uncovered to have lower serum zinc levels, which were found for adults and children [125,126]. Direct effects on clinical symptoms including nasal irritation, rhinorrhea, and facial pain, as well as mucosal thickening, polypoid changes, and eosinophilia, could not be shown by the above-mentioned studies. However, two studies by Akbari et al. and Dewi et al. highlighted that oral zinc supplementation significantly improves the clinical status and general health of CRS patients [72,127]. Moreover, zinc levels in both epithelium and stroma were demonstrated to be decreased in CRSsNP and CRSwNP patients compared to healthy controls [128]. However, no significant differences in serum zinc levels between CRSsNP and CRSwNP patients could be discovered [129]. Existing data seem to be still controversial, since Murphy et al. only uncovered that zinc levels in nasal mucosa were significantly decreased in CRS patients compared to healthy controls, but serum zinc levels remained equal in all groups [129]. They uncovered a trend in serum zinc levels in CRSsNP, however, which were lower than those in CRSwNP patients, but differences did not reach statistical significance.

One possible molecular mechanism for altered intracellular zinc levels might be the altered expression of the zinc transporter of the nasal mucosa in patients suffering from CRS [129]. Altered expression patterns are described for CRSsNP patients but not for CRSwNP patients [128,129]. Hence, more research seems to be needed to find an explanation as to why serum zinc levels are still low in both patient groups. In CRSsNP patients, Murphy et al. found that the zinc importers (ZIP)1, ZIP2, ZIP14 are significantly decreased, as well as the intracellular zinc chelators, metallothionein (MT)1a, MT2a, and MT3, whereas the zinc exporter (ZnT)1 is significantly increased. In vitro studies confirmed those findings by showing zinc supplementation normalized the zinc-transporter expression in airway epithelia (see Figure 1), whereas inflammation was shown to have the opposite effect [130,131]. At the side of acute inflammation, increased mucus-zinc levels can be found, but not in mucus harvested from unaffected sides of the same patients [132]. Hence, a local increased zinc release via nasal discharge at sites of acute inflammation could be one possible explanation for lower intracellular zinc levels. Mucosal zinc depletion might play a central role in CRS pathogenesis, since epithelium barrier dysfunction and tissue remodeling are highly dependent on adequate zinc levels [133,134].

Patients with CRS often show an aberrant and impaired nasal epithelial barrier function [133,134,135,136]. The epithelium barrier consists of intercellular junctional complexes between neighboring cells, which provide continuous cell–cell contact to protect the human body from invading pathogens and uncontrolled absorption or release of substances. These complexes are composed of several units, including the tight junctions (TJs) and adherens junctions (AJs), which form circumferential zones of contact between adjacent cells. The main transmembrane adhesion molecule E-cadherin is localized at the AJ, which is responsible for appropriate AJ organization. Tight junction component proteins like claudin1, occludin, and zonula occlundes1 are decreased in sino-nasal epithelia of people suffering from CRS [135,136]. In line with this, zinc deficiency was shown to accelerate the proteolysis of E-cadherin and β-catenin, leading to increased leakage across the monolayer of upper and alveolar lung epithelial cultures [87,137]. Consequently, uncontrolled neutrophil migration through the disrupted junctional complexes is triggered, which leads to exacerbated inflammation, reinforcing mucosal damage [138]. In contrast, zinc supplementation ameliorates lung injury by reducing neutrophil migration and activity [139]. Moreover, zinc supplementation beneficially affects the overall barrier function by restoring membrane function, improving tight junction stability, and reducing cell apoptosis [138,140]. In line with this, a clinical study uncovered that serum zinc levels were significantly increased by oral zinc supplementation for 6 weeks, and the general health of patients with CRS was improved [72].

## 4. Zinc and Allergic Gastrointestinal Disorders

Nutrition plays a critical role in shaping the gut microbiota, which is essential for maintaining the integrity of the intestinal epithelial barrier and intestinal immune homeostasis. In addition, nutrients and their endogenous or bacterial metabolites can trigger allergic inflammation in distant organs outside the gut, such as the lungs (via the gut–lung axis) and skin (via the gut–skin axis) [67], making gastrointestinal health essential for the overall health of the human body.

The prevalence of food allergies has increased dramatically over the last four decades. Worldwide, more than 10% of the general population is affected [141]. This is an alarming clinical problem, especially in younger children, as normal development may be impaired [42]. One in twelve children is affected by a food allergy, with the highest levels occurring in the first three years of life [142]. Nearly 33 million Americans have an IgE-mediated food allergy, and more than 50% of adults and 42% of children have experienced a severe reaction [143]. Around 90% of food allergies are caused by the eight most common allergens (milk, eggs, nuts, peanuts, fish, crustaceans, wheat, and soybeans) [144]. Persistent exposure to allergens can lead to chronic inflammation, which alters the protective mucosa and increases reactive oxygen species (ROS) production. Disturbances of this barrier function lead to many unfavorable reactions, including clinical symptoms such as diarrhea or flatulence, or, at the molecular level, oxidation of cell membrane lipids and impaired function [144]. During zinc deficiency, ROS production is significantly increased; this might be due to complex interplay of altered gene expression and enzymatic activity, which is summarized in detail elsewhere [145]. Briefly, zinc affects the redox metabolism by affecting the superoxide dismutase activity, metallothionein (MT) activity, and expression and formation of intramolecular di-sulfide bonds, and modulates mitochondria and endoplasmic reticulum function [145].

MT activity is known to modulate caspase inactivation. Hence, a negative correlation in MT and caspase activity is known. Since MTs are inducible by various stimuli, like proinflammatory cytokines, glucocorticoids, oxidative stress, and cations of divalent metals, such as zinc [146,147], zinc supplementation leads to reduced cell apoptosis by MT-triggered caspase inactivation, and scavenged and limited ROS production, which protects cells from DNA damage [148,149].

Zinc supplementation is known to maintain the oxidative–antioxidative balance by influencing ROS production via the activity and stability of superoxide dismutase [150]. A clinical study showed that children with IgE-mediated and IgE-independent food allergies had significantly lower serum zinc levels and thus a weakened antioxidant barrier compared to healthy children [150]. At the molecular level, studies showed zinc to be important for mucosal barrier morphology and function, by regulation of claudin-3 and occludin expression (see Figure 1), since zinc deficiency injured the intestinal mucosal barrier by decreasing expression of the tight junction proteins, increasing intestinal epithelial permeability, impairing intestinal mucosal barrier function, and reducing cell viability [151,152]. The repair and regeneration of the intestinal mucosa can be triggered by exogenous zinc supplementation, leading to restored integrity of the intestinal mucosal barrier morphology and function [153]. At the molecular level, zinc administration activates the membrane-bound G protein-coupled receptor (GPR) 39, which mainly triggers the intracellular rise of calcium concentrations through a Gαq-phospholipase C (PLC)-inositol triphosphate (IP3) pathway. Eventually, the PLC-Ca(2^+^)/calmodulin-dependent protein kinase kinase-β (CaMKKβ)-serine/threonine kinase AMP-activated protein kinase (AMPK)-pathway-dependent tight junction abundance is raised. Hence, epithelial integrity is improved [154,155].

Of the children studied with food allergies, significantly more were artificially fed from birth. As the intestinal absorption of zinc from breast milk is considerably higher, artificial feeding is more likely to lead to zinc deficiency and to promote food allergies [150]. In line with this, children with allergic proctocolitis had lower zinc levels than healthy controls (see Table 1). However, no correlation was found between zinc levels and the time of onset of symptoms [156].

Nevertheless, it is important to remember that the modern diet has changed in recent decades and that the diet itself could cause some health problems. Greater environmental and health awareness has motivated many people to change their eating habits and avoid animal products and adopt a vegetarian or vegan diet. However, due to the lower bioavailability of zinc in plant-based foods because of non-digestible plant ligands, like phytate and lignin, zinc is chelated, and intestinal absorption is aggravated. Absorption is furthermore impaired by calcium and iron content of the food, making zinc absorption even more complex [157,158]. Hence, zinc deficiency can occur more frequently in vegetarian or vegan diet, as zinc requirements can be up to 1.5 times higher [159]. It is therefore important, especially for children, to ensure a balanced diet to guarantee a sufficient zinc status and thus to ensure health of the gastrointestinal tract. This is why both the World Health Organization (WHO) and the European Food Safety Authority (EFSA) consider the inhibitory effect of dietary phytate on zinc absorption in the intestine when setting the recommended zinc intake values [160,161].

The WHO categorizes diets into three groups regarding zinc absorption efficiency in the small intestine: A (high, <5), B (moderate, 5–15), and C (low, >15). This classification is based on the molar phytate–zinc ratio of the diet ranging from below 5 to up to 15, which strongly influences zinc absorption. In the high bioavailability class A (high, <5), zinc is highly available in the diet, and the intake that promotes the absorption of 1 mg zinc/day must be about 1.8 mg zinc, which is a realistic utilization efficiency of 56%. In category B, zinc is moderately available, and the diet has a molar phytate–zinc ratio ranging between 5 and 15. Diets rich in antagonists that interfere with zinc absorption are summarized in category C (low, >15). They show a phytate–zinc ratio of above 15 and are indicated to have a zinc availability of merely 10–15% [160].

The EFSA calculates estimated average requirements and Population Reference Intakes of zinc for phytate intake levels of 300, 600, 900, and 1200 mg/day, which cover the median intakes observed in the European population [161]. It is therefore important to pay attention to the exact composition of the diet for adequate zinc absorption to prevent malnutrition.

## 5. Zinc and Allergic Skin Disorders

The skin is the largest organ of the human body and is one of the most important barriers to the environment. A lot of people suffer from an inadequate barrier function, which leads to various diseases like atopic dermatitis. Atopic dermatitis is a chronic, relapsing inflammatory disorder characterized by intense itching and eczematous skin lesions affecting 10–20% of the population in Western countries. The prevalence is also increasing in developing countries [162].

Disruption of the skin barrier has been shown to be a major reason for the development of atopic dermatitis [162,163,164], as has infiltration of the skin by inflammatory immune cells like Th2 cells. Similar mechanisms of inflammatory lymphocyte infiltration in the skin are also known for alopecia areata. As with atopic dermatitis, the exact causes of the diseases are not yet fully understood, but it is currently assumed that a complex interplay between environmental and genetic factors and nutrition status might be responsible, as altered micronutrient levels like zinc have been found in these patients [165].

The skin contains the third highest amount of zinc in the human organism. The zinc distribution in the skin decreases from the outer to the inner area [166]. The epidermis contains more zinc than the dermis, and within the epidermis, zinc is mainly found in the stratum spinosum compared to the other layers of keratinocytes [167,168]. In the dermis, there is also more zinc in the upper part than in the lower part. Zinc acts as a stabilizer of cellular membranes in epidermal and dermal tissues, as shown in Figure 1, and is an essential co-factor of enzymes like matrix metalloproteinases. These play a critical role in wound healing by enhancing auto-debridement and keratinocyte migration and function [169]. Additionally, zinc enhances epithelial resistance against ROS and toxins via the antioxidant activity of metallothionein (see Figure 1), thus preventing cellular apoptosis [170].

During skin injury, extracellular zinc is released; this causes activation of the G-protein coupled receptor (GPR39) and the zinc-sensing receptor (ZnR), which are expressed in keratinocytes and other epithelial cells, respectively. This assists epithelium repair and dampens the inflammatory immune response [171,172,173]. Hence, zinc is widely used in formulating cosmetics and ointments to support wound healing and to force antimicrobial activities to facilitate wound healing [174,175,176].

Zinc deficiency, on the other hand, contributes to delayed wound healing, which is mainly seen in the elderly population having an impaired nutritional status [170,177]. In cultured keratinocytes, zinc deficiency causes caspase-3 activation and apoptosis induction [178]. Furthermore, keratinocyte proliferation, differentiation, and survival are negatively affected, leading to poor wound healing, increased ATP and inflammatory cytokine production, and altered inducible-NO synthase (iNOS)/NO synthesis, which provokes abnormal hair keratinization and temporal hair loss [168,179].

For decades, zinc deficiency has been known to play an essential role in development of several skin diseases, as first noted in the 1970s for acrodermatitis enteropathica [42]. Therefore, it is not surprising that zinc supplementation is widely used in various skin diseases, including infectious diseases such as viral warts and genital herpes, inflammatory diseases such as acne vulgaris, psoriasis, pigmentary diseases such as vitiligo and melasma, tumor-related diseases such as basal cell carcinoma, endocrine, metabolic diseases such as necrotic erythema, and hair diseases such as alopecia [112,168].

However, data on zinc deficiency and its influence on atopic dermatitis are still controversial (see Table 1). Some studies proclaimed no differences in serum zinc levels of atopic dermatitis patients compared to healthy individuals [50,51,52,180,181,182] and that the severity of atopic dermatitis does not depend on serum zinc levels [181]. Other studies found that patients with atopic dermatitis had lower serum zinc levels compared to healthy controls [183,184,185,186] and showed that patients with advanced forms of atopic dermatitis had even lower serum zinc levels compared to patients with milder forms [106,187]. Other studies also found significantly lower zinc levels in the erythrocytes or hair of children and adults with atopic eczema compared to healthy individuals [106,111,182,184,185,186,188], as well as low zinc transporter expression (ZIP10) and decreased zinc-dependent enzymes in atopic lesions [189,190]. In addition, a negative correlation was found between the zinc content in erythrocytes and the scoring index for atopic dermatitis (SCORAD) [182,184]. In line with this, similar results were found in vitro, showing decreased skin water content, increased trans-epidermal water loss (TEWL), and increased serum IgE levels [191].

Research on the efficacy of zinc supplementation is also still controversial, as some studies have found no effects of zinc supplementation during pregnancy and the development of atopy in the offspring [48,52]. Other studies have found positive effects of oral zinc supplementation, as evidenced by increased serum zinc levels and a decreased Eczema Assessment Severity Index (EASI), decreased TEWL, lower scores on visual analog scales for pruritus, and sleep disturbance [188] (see Table 2).

Comparable results are found for the inflammatory skin disease alopecia areata. Serum zinc levels are certainly decreased in patients with severe alopecia areata, showing a prolonged progression and resistance to conventional therapies [192,193,194,195,196,197,198]. Conversely, another study showed only one of twenty-two patients to be zinc deficient [199]. Two studies uncovered additional zinc supplementation to beneficially influence alopecia areata, showing accelerated hair regrowth in patients with alopecia areata compared to untreated individuals [200,201]. One study showed no significant difference [202].

Nevertheless, zinc administration to humans is reported to be safe, having no significant side effects observed so far [203,204]. Since several studies found a significant zinc deficiency in patients suffering from allergic diseases, zinc supplementation should be considered as a promising additional treatment option for allergic diseases [66,89,205,206,207].

## 6. Conclusions

Taken together, a large number of clinical studies link low serum, erythrocyte, sputum, hair, and nail zinc levels to allergic diseases (see Table 1); however, there are still contradictory results existing that could be due to different study designs. Nevertheless, the above-described effects of zinc supplementation underscore the fundamental role of the micronutrient zinc in a healthy immune response and in maintaining the structural integrity of the barrier sites of the human body. There seems to be great potential to improve the use of zinc as a therapeutic agent in various diseases, as highlighted in this article for allergic asthma, CRS, atopic dermatitis, alopecia areata, food allergies, and allergic proctocolitis (see Table 2). However, the interactions of zinc with other nutritional elements need to be examined further to develop supplementation measures that target multiple deficiencies to successfully support disease treatment. Nevertheless, the general expert recommendations by the WHO, National Institute of Health (NIH), or the German Nutrition Society (DGE) should be followed in the case of supplementation strategies. Interactions with other nutrients influencing their absorption and utilization are described at total zinc intakes as low as 60 mg/day, and the mean intake should not exceed 45 mg/day [160]. The studies summarized in Table 2 follow this general WHO recommendation. The recommended daily amount regarding the National Institute of Health is 11 mg/day for adult men, 8 mg/day for adult women, and 11–12 mg/day during pregnancy and lactation [208]. The DGE recommends 11–16 mg/day for adult men and 7–10 mg/day for adult women, depending on phytate intake. During pregnancy and lactation, up to 13–14 mg/day is recommended. For infants and children, the RDA is 2.5 mg/day for both sexes [209]. Due to the inhibitory effect of phytate on zinc absorption from food, phytate levels are taken into account by both the WHO and the EFSA when establishing recommendations for daily zinc intake [160,161].

However, further research is also necessary to establish whether zinc deficiency is a risk factor for the severity and outcome of allergic diseases. Although oral zinc supplementation is known to be safe and economical, research is needed to define precise and uniform treatment strategies for allergic diseases.

**Table 1 biomolecules-14-00863-t001:** Zinc status in allergic patients.

Disease	Participation	Age	Results/Symptoms	Reference
Asthma	50 (D), 50 (C)	2–18 yr	Significant lower serum zinc levels compared to healthy controls	[110]
	50 (D), 50 (C)	1–12 yr		[113]
	49 (D), 24 (C)	10–50 yr		[97]
	24 (D), 24 (C)	8–18 yr		[118]
	36 (D), 36 (C)	10–30 yr		[99]
	50 (D), 70 (C)	5–18 yr		[121]
	100 (D), 100 (C)	6 yr		[103]
	40 (D), 43 (C)	18–70 yr		[100]
	22 (D), 33 (C)	-		[104]
	554 (D), 1312 (C)	30–60 yr		[89]
	73 (D), 75 (C)	3–24 mon		[95]
	34 (D), 14 (C)	Infants		[101]
	80 (D), 80 (C)	2–15 yr		[107]
	6 (D), 12 (C)	6–12 yr		[114]
	51 (D), 541 (C)	6–12 yr		[102]
	52 (D), 38 (C)	25–48 yr		[96]
	60 (D), 30 (C)	38–52 yr	Significantly lower serum zinc levels and significantly higher IgE levels and worse FEV1 in asthmatic patients	[55]
	46 (D), 30 (C)	20–65 yr		[91]
	25 (D), 25 (C)	30–40 yr		[210]
	71 (D), 0 (C)	7–17 yr		[105]
	114 (D), 49 (C)	41–71 yr	Significantly lower sputum zinc levels compared to healthy controls	[98]
	40 (D), 20 (C)	2–12 yr	Significantly decreased nail and hair zinc levels compared to healthy controls	[183]
	22 (D), 19 (C)	2–14 yr		[111]
	34 (D), 14 (C)	1–3 yr		[101]
	40 (D), 40 (C)	7–14 yr	No difference in serum zinc levels in patients compared to healthy controls	[117]
	42 (D), 30 (C)	2–14 yr		[74]
	175 (D), 165 (C)	3–19 yr		[93]
	80 (D), 80 (C)	3–9 yr		[120]
	46 (D), 43 (C)	3 mon–2 yr		[95]
	30 (D), 30 (C)	mean age 41		[122]
	19 (D), 17 (C)	above 18 yr		[115]
	100 (D), 170 (C)	20–65 yr	Significantly elevated serum zinc levels in patients vs. healthy controls	[123]
	67 (D), 45 (C)	below 18 yr	No difference in erythrocyte zinc levels compared to healthy controls and no relationship between zinc levels and duration of follow up, severity, and control of asthma	[116]
	37 (D), 30 (C)	8–18 yr	No effect beween serum zinc level and serum IgE levels or Skin Test Reactivity	[119]
CRS	28 (D), 7 (C)	above 18 yr	Significantly reduced zinc level in biopsy of nasal epithelium	[129]
	28 (D), 8 (C)	32–44 yr	Significantly reduced tissue zinc levels in correlation with a reduction in collagen content, and increased eosinophil numbers	[128]
	24 (D), 20 (C)	7–12 yr	Significantly decreased serum zinc levels compared to healthy controls	[125]
Atopic dermatitis	42 (D), 126 (C)	3 yr	Zinc deficiency significantly correlates with AD severity and elevated serum IgE levels	[106]
	67 (D), 49 (C)	9–27 yr	Significantly lower erythrocyte zinc levels in AD patients compared to healthy controls, negative correlation between the SCORAD score and erythrocyte zinc levels	[182]
	92 (D), 70 (C)	2–4 mon	Erythrocyte zinc levels were significantly lower in AD patients compared to healthy controls	[184]
	58 (D), 43 (C)	2–14 yr	Significantly decreased hair zinc levels, but no alteration of serum zinc levels in AD patients and healthy controls	[188]
	65 (D), 79 (C)		Significantly reduced serum zinc levels in AD patients compared to healthy controls, and recurrent infections of the skin	[186]
	105 (D), 105 (C)	1–12 yr	Significant difference in median zinc between children with AD and healthy controls	[185]
	43 (D), 19 (C)	2–14 yr		[111]
	20 (D), 20 (C)	5–12 yr	Significantly lower serum zinc levels in patients with moderate AD compared to patients with mild AD, negative correlation between serum zinc levels and severity of AD	[187]
	18 (D), 20 (C)	-	Significantly lower serum zinc levels and hair zinc levels compared to healthy controls	[183]
	134 (D), 112 (C)	-	No difference in serum zinc levels in patients vs. healthy controls	[180]
	160 (D), 79 (C)	-		[181]
Alopecia areata	49 (D), 32 (C)	-	Significantly lower serum zinc and hair zinc levels compared to healthy controls	[198]
	32 (D), 32 (C)	5–31.5 yr		[168]
	50 (D), 50 (C)	17.5–36.5 yr	Significantly lower serum zinc levels compared to healthy controls	[194]
	50 (D), 50 (C)	27 yr		[193]
	77 (D), 112 (C)	16–43 yr		[195]
	60 (D), 60 (C)	20–55 yr		[196]
	30 (D), 30 (C)	19–48 yr		[197]
Food allergy	134 (D), 36 (C)	1–36 mon	Significantly lower serum zinc levels compared to healthy controls	[150]
	50 (D), 50 (C)	4–10 yr	Significantly lower intracellular zinc levels in erythrocytes in patients with FPIAP compared to healthy controls	[156]

D: people suffering from allergic disease, C: healthy controls, AD: atopic dermatitis, IgE: immunoglobulin E, FEV1: forced expiratory volume in one second, SCORAD: scoring index for atopic dermatitis, FPIAP: food protein-induced allergic proctocolitis.

**Table 2 biomolecules-14-00863-t002:** Zinc supplementation in allergic diseases.

Disease	Participation	Zinc Supplementation	Symptoms/Effects	Reference
Asthma	144 (I), 140 (C)	50 mg daily, 8 weeks	Elevated serum zinc level, improvement in clinical symptoms	[109]
	21 (I), 21 (C)	30mg daily (ZB), 4 days	Decreased severity of asthma in the first 48 hours after admission	[108]
	797 women	12.5 mg daily during pregnancy	Significantly lower appearance of asthma events and asthma activity	[94]
CRS	28 (I), 16 (C)	55 mg elemental zinc, 6 weeks	Significant improvement in clinical status and general health	[72]
	34 (I), 0 (C)	40 mg elemental zinc, 2 weeks	Significant reduction in mean total symptom score and improvement in mean quality of life score after supplementation	[127]
Atopic dermatitis	12 (I → C)	ZO-textiles (trousers and long-sleeve shirts)	Less pruritus, improvement in sleep quality and clinical cutaneous symptoms	[211]
	797 women	12.5 mg daily, during pregnancy	Significantly reduced appearance of eczema, doctor-confirmed eczema, and less intense treatment	[94]
	58 (I), 43 (C)	12 mg daily (ZO), 8 weeks	Significantly increased hair zinc levels and decreased EASI, TEWL, visual analogue scales for pruritus, and sleep disturbance	[188]
	420 pregnant women, 300 children	21 mg daily, during pregnancy	No relationship between zinc supplementation during pregnancy and allergic outcome in 1-year-old children	[50]
	1002 pregnant women	8.5 mg daily, during pregnancy	No association between zinc intake and allergic rhinitis	[51]
	763 mother-child pairs	8.5 mg daily, during pregnancy	No association between zinc intake and risk of wheeze or eczema in the children	[52]
Alopecia areata	-	100 mg daily (ZA), 20 days	Less clinical cutaneous symptoms, no statistical differences between treatments in term of eyebrow regrowth	[200]
	100 (I), 100 (C)	1% pyrithione zinc shampoo, 9 weeks	Significantly elevated hair counts	[212]
	15 (I → C)	50 mg daily (ZG), 12 weeks	Significantly elevated serum zinc levels, no statistically significant hair regrowth	[192]
	37 (I), 37 (C)	5 mg/kg/d (ZS), 3 months	Complete hair regrowth after 2 months of intervention compared to placebo	[201]
	21 (I), 21 (C)	220 mg daily (ZS), 3 months	No improvement in extent or activity of diseases but slight raise in serum zinc and hair zinc levels compared to healthy controls	[202]

I: people treated with zinc (Intervention), C: untreated group, I → C: people first treated with zinc and then re-examined as untreated controls, ZB: zinc bis-glycinate, ZO: zinc oxide, ZA: zinc aspartate, ZG: zinc gluconate, ZS: zinc sulphate, C-ACT: childhood asthma control test, EASI: eczema assessment severity index, TEWL: trans-epidermal water loss.

## Figures and Tables

**Figure 1 biomolecules-14-00863-f001:**
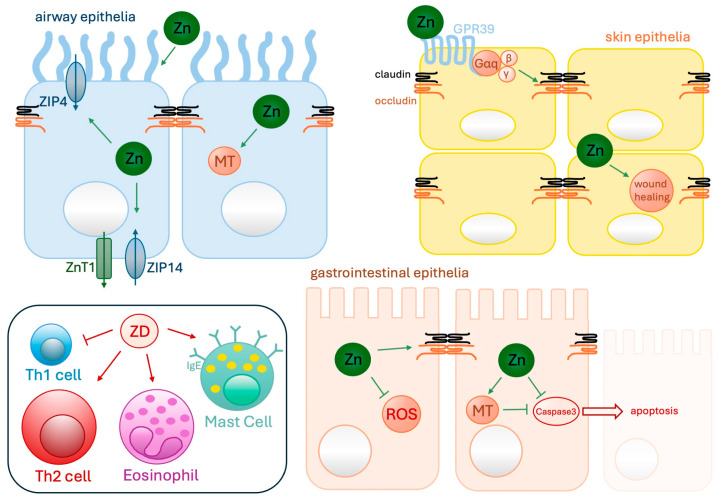
Zinc and cellular processes in human airway epithelia, skin epithelia, and gastrointestinal epithelia. Zinc supplementation normalizes zinc-transporter (ZnT1, ZIP4/14) expression and increases metallothionein (MT) expression. Tight junction stability is enhanced by activation of the G-protein coupled receptor GPR39 (see skin epithelia; Gαq-receptor; Gαq: G protein alpha q subunit), leading to increased occludin (orange transmembrane protein) and claudin (black transmembrane protein) stability, which assist epithelium repair. Cellular resistance against apoptosis by inhibiting caspase-3 activity is forced, and toxin-induced reactive oxygen species (ROS) production is reduced due to the antioxidant activity of metallothionein, which dampens the inflammatory immune response. Zinc deficiency (ZD) triggers differentiation of mast cells within the bone marrow and mast cells contain increased numbers of specific granules. Moreover, the T helper (Th)1/Th2 ratio is imbalanced by zinc deficiency, showing a reduction in Th1 cells. Thus, Th2 cell responses are triggered, leading to elevated eosinophil counts and altered humoral immunity by activating B cells and antibody production, which is mainly involved in allergic diseases.

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
