# Peer review of "Zinc Deficiency and Zinc Supplementation in Allergic Diseases"

_biomolecules, 2024, doi:10.3390/biom14070863_

Round 1

Reviewer 1 Report

Comments and Suggestions for Authors

Here my suggestions to improve your work:

1- line 51, please cite some data on the prevalence of asthma and atopic disease in an Asian country (e.g. Japan or another)

2- line 54, ' altered population diet' might be the Westernization of dietary patterns. If so correct the sentence.

3-line 56 Cooms and Gell classified the hypersensitivity reactions, not the allergic immune response, please change your sentence.

4-Line 80, please substitute the term microflora (old term) with the actual used term microbiota.

5-Lines 119-124 please insert even the Iron supplementation for a long time. indeed excessive load of iron can impact the zinc absorption in the GI tract.

6-lines 354-356 explore the WHO categorization with more sentences.

7-Table 2 please insert a description with the acronym used in Table 2 like Table 1

8-Conclusion: set the practical tips for supplementation of Zinc in allergies with dose and duration, and resume the data of Table 2.

Comments on the Quality of English Language

None

Author Response

We thank REVIEWER 1 for the helpful comments.

Here my suggestions to improve your work:

1- line 51, please cite some data on the prevalence of asthma and atopic disease in an Asian country (e.g. Japan or another)

Thank you very much for this note. Corresponding literature references are inserted (see line 51).

2- line 54, ' altered population diet' might be the Westernization of dietary patterns. If so correct the sentence.

 Thank you very much, the mentioned sentence is corrected (see lines 56,57).

3-line 56 Cooms and Gell classified the hypersensitivity reactions, not the allergic immune response, please change your sentence.

This is a very useful advice. It has been corrected that the classification is mainly relevant for hypersensitivity reactions (see line 62, highlighted in red).

4-Line 80, please substitute the term microflora (old term) with the actual used term microbiota.

 The actual term microbiota is now used (see line 87, highlighted in red).

5-Lines 119-124 please insert even the Iron supplementation for a long time. indeed excessive load of iron can impact the zinc absorption in the GI tract.

Thank you, this is a very good point. In the text it is added (see lines 129-130, highlighted in red).  that simultaneous supplementation of high amounts of iron negatively influences zinc absorption especially if administered in liquid form, which was described for example by Solomons et al., Sandstroms et al., and Harvey et al.

In fasting human subjects, a dose-dependent decrease in zinc absorption as measured by area under the curves during 4 h time period post-iron-dosing at 2–3-fold molar excess has been uncovered (Solomons et al.) The inhibitory effect of iron on zinc absorption was also found to be higher with ferrous iron than its ferric counterpart (Sandstrom et al.). Together, these studies suggest competitive interaction between iron and zinc during intestinal absorption, especially when high iron doses are added. In agreement, negative interaction of iron on zinc absorption are found when administered in liquid form (Harvey et al.).

6-lines 354-356 explore the WHO categorization with more sentences.

Indeed, it makes sense to describe die WHO categorization in more detail. Thank you very much for your advice. The changes are now inserted in lines 383-392 (highlighted in red).

The classification is based on the molar phytate-zinc-ratio of the diet ranging from below 5 to up to 15, which strongly influences zinc absorption. In the high bioavailability class A (high, <5) zinc is highly available in the diet and the intake that promotes the absorption of 1 mg zinc/day must be about 1.8 mg zinc, which is a realistic utilization efficiency of 56%. In category B, zinc is moderately available, and the diet has a molar phytate-zinc ratio ranging between 5 and 15. Diets rich in antagonists that interfere zinc absorption are summarized in category C (low, >15). They show a phytate-zinc-ratio of above 15 and are indicated to have a zinc availability of merely 10-15%

7-Table 2 please insert a description with the acronym used in Table 2 like Table 1

 The explanations of the acronyms are now inserted as a footnote as in Table 1.

8-Conclusion: set the practical tips for supplementation of Zinc in allergies with dose and duration, and resume the data of Table 2.

Thank you very much for this suggestion. Changes are inserted in lines 486-500 (highlighted in red).

It is not trivial to give a simple overall recommendation for an adequate zinc intake for the population worldwide, as the bioavailability of zinc varies greatly depending on the carrier material. Food also plays an important role, as zinc chelators such as phytates significantly limit zinc absorption and thus increasing its loss in faeces. Moreover, different population groups such as children or pregnant women require a higher zinc intake to meet the requirements for basal zinc level.

Regarding the WHO, the Expert Consultation recommended that the adult population mean intake should not exceed 45 mg. Interactions with other nutrients influencing their absorption and utilization have been detected biochemically at total zinc intakes as low as 60 mg/day. Studies summarized in Table 2 follow this general WHO recommendation. The recommended daily amount regarding the National Institute of Health is 11 mg/day for adult men, 8 mg/day for adult women and during pregnancy and lactation 11-12 mg/day (NIH 2024). The German Nutrition Society (DGE) recommends 11–16 mg/day for adult men and 7–10 mg/day for adult women, depending on phythate intake. During pregnancy and lactation up to 13–14 mg/day are recommended. For infants and children, the RDA is 2.5 mg/day for both sexes (DGE 2024). Due to the inhibitory effect of phytate on zinc absorption from food, phytate levels are taken into account by both the World Health Organization (WHO) and the European Food Safety Authority (EFSA) when establishing recommendations for daily zinc intake (WHO, EFSA).

References

Solomons, N.W.; Jacob, R.A. Studies on the bioavailability of zinc in humans: Effects of heme and nonheme iron on the absorption of zinc. Am. J. Clin. Nutr. 1981, 34, 475–482.

Sandstrom, B.; Davidsson, L.; Cederblad, A.; Lonnerdal, B. Oral iron, dietary ligands and zinc absorption. J. Nutr. 1985, 115, 411–414.

Harvey, L.J.; Dainty, J.R.; Hollands, W.J.; Bull, V.J.; Hoogewerff, J.A.; Foxall, R.J.; McAnena, L.; Strain, J.J.; Fairweather-Tait, S.J. Effect of high-dose iron supplements on fractional zinc absorption and status in pregnant women. Am. J. Clin. Nutr. 2007, 85, 131–136.

Reviewer 2 Report

Comments and Suggestions for Authors

Reviewer’s comments

This review article written by Maywald et al. primarily focuses on the allergic reactions induced by zinc deficiency. Moreover, the effects of zinc supplementation on several allergic diseases have been also discussed in this article. However, the putative mechanisms by which zinc deficiency causes allergic reactions have not been fully mentioned yet. The usefulness of zinc supplementation on the allergic diseases has not been appropriately evaluated, using several biochemical, physiological or immunological parameters. Several revisions are required for the better quality of this review article. Please refer to the comments shown below.

Major

#1. The title of this article does not seem to be impressive to the readers. The keywords that reflect this article have not been appropriately selected.

#2. The purpose of this study remains uncertain. The authors should clearly describe it in “Introduction”.

#3. The putative mechanisms by which zinc deficiency evokes asthma have not been fully described. The statement on the correlation between zinc deficiency and eosinophilia would be also required in such allergic diseases. Therefore, the authors should revise the Figure 1 (zinc deficiency) to explain the cellular immunity including the role of eosinophils in the pathogenesis of asthma.

#4. The authors should describe the effects of zinc supplementation on asthma more clearly. For instance, the statement that zinc supplementation improved lung parameters such as FVC, FEV1 and FEV1/FVC is found (Page 5, lines233-235). However, the findings are not shown in Table 2. Moreover, the references that zinc supplementation contributed to asthma prevention and symptom improvement are cited [77, 78, 90-98] (Page 5, line 235-236). However, the references #78, #90, #94-#98 are missing in Table 2. The authors should verify the efficacy of zinc supplementation using biochemical or physiological parameters. The alteration of IgE and the numbers of eosinophils by zinc supplementation should be explored.

#5. The explanation for Zn-transporters is missing in Figure 1. The authors should note which indicate GPR39 and claudin-3 respectively in Figure 1 (skin epithelia). The explanation for this figure seems to be inadequate. What is Gaq in the figure? The role of metallothionein in the inhibition of caspase-3 activity should be also illustrated in Figure 1 (gastrointestinal epithelia). The authors should cite the reference that zinc supplementation inhibits the activity of caspase-3 by way of the induction of metallothionein.

#6. The section for CRS (Page 6 line 251-Page 7 line 297) should have a new subtitle, “Zinc and Chronic Rhinosinusitis”. The correlation between the zinc status and the presence or absence of nasal polyp has not been clearly described in the section. Is the effect of zinc supplementation approximately equivalent between CRSwNP and CRSsNP? Moreover, the figure illustrating the role of zinc in the nasal epithelium is missing.

#7. How does GPR39 affect impaired tight junctions by zinc supplementation? The authors should describe a putative mechanism by which GPR39 is involved in the repairment of tight junctions.

#8. The statements on conflict of interest, authors’ contribution, external fund and acknowledgement are missing. These are required items for submitting an article.

Minor

#1. ZA, ZB, ZG and ZO should be spelled out in Table 2, respectively.

#2. What indicates (I) in Table 2? What does Wiegand mean in Table 2?

#3. The abbreviations for Journal can be used in “References”. For instance, Allergy International can be abbreviated as Allergol Int (Reference 116).

Comments on the Quality of English Language

Several editions are required.

Author Response

We thank REVIEWER 2 for the helpful comments.

Reviewer’s comments

This review article written by Maywald et al. primarily focuses on the allergic reactions induced by zinc deficiency. Moreover, the effects of zinc supplementation on several allergic diseases have been also discussed in this article. However, the putative mechanisms by which zinc deficiency causes allergic reactions have not been fully mentioned yet. The usefulness of zinc supplementation on the allergic diseases has not been appropriately evaluated, using several biochemical, physiological or immunological parameters. Several revisions are required for the better quality of this review article. Please refer to the comments shown below.

Major

#1. The title of this article does not seem to be impressive to the readers. The keywords that reflect this article have not been appropriately selected.

We thank the reviewer for this note and changed the title to “Zinc deficiency and zinc supplementation in allergic diseases”, which described more clearly its content. The keywords are exhaustive for the content of the review.

#2. The purpose of this study remains uncertain. The authors should clearly describe it in “Introduction”.

 Thank you so much for your notice. A corresponding passage has been inserted in the paragraph “Introduction” lines 49-60 (highlighted in red).

#3. The putative mechanisms by which zinc deficiency evokes asthma have not been fully described. The statement on the correlation between zinc deficiency and eosinophilia would be also required in such allergic diseases. Therefore, the authors should revise the Figure 1 (zinc deficiency) to explain the cellular immunity including the role of eosinophils in the pathogenesis of asthma.

Thank you very much for this point. Three mechanisms of the influence of zinc deficiency are already described in this review: Zinc deficiency disturbs the balance between type 1 and type 2 T helpers, resulting in inflammation and eosinophilia. Eosinophilia/Eosinophil cells are now mentioned in lines 150-157, 217, 222, 234/235 and 258/259, 279. New aspects added to the text are highlighted in red. Additionally, zinc deficiency may be associated with the production of IgE, which increases the risk of asthma. Moreover, zinc is of powerful antioxidant activity in the lungs, which may be responsible for the imbalance between oxidation and antioxidant, ultimately reducing antioxidant function and increasing the risk of asthma. (Xue et al.)

Allergic asthma is generally characterized by increased production of type 2 cytokines and accumulation of lymphocytes and eosinophils into the airway. Herein, Group 2 innate lymphoid cells (ILC2s) have been identified as critical upstream mediators of type 2 inflammation and the induced airway eosinophilia in translational models of allergic asthma (Bartemes et al. 2014, Bartemes et al. 2021, Doherty et al.). Although ILC2s are proposed to be upstream mediators of eosinophils, little is known regarding the reciprocal interactions between eosinophils and ILC2s despite their co-occurrence in allergic diseases (LeSuer). This aspect has been added in lines 150-157 (highlighted in red).

Studies uncovered, that there seem to be only a weak correlation at baseline between eosinophilic inflammation and bronchial hyper-responsiveness (Crimi et al., Wislon et al.). The anti-IgE monoclonal omalizumab reduces airway eosinophilia but has no effect on bronchial hyper-responsiveness (Djukanović et al.), whereas the anti-TNF monoclonal etanercept reduces hyper-responsiveness but has no effect on airway inflammation (Berry et al.). Furthermore, there is no relationship between the extent of airway remodeling, specifically reticular basement membrane thickness, and the degree or duration of any inflammatory parameter (Payne et al.). Hence, the role of eosinophilic inflammation and the severity of the disease therefore appears to require further research and details will not be added to the review.

The role of eosinophils in zinc deficiency is now added to Figure 1.

#4. The authors should describe the effects of zinc supplementation on asthma more clearly. For instance, the statement that zinc supplementation improved lung parameters such as FVC, FEV1 and FEV1/FVC is found (Page 5, lines233-235). However, the findings are not shown in Table 2. Moreover, the references that zinc supplementation contributed to asthma prevention and symptom improvement are cited [77, 78, 90-98] (Page 5, line 235-236). However, the references #78, #90, #94-#98 are missing in Table 2. The authors should verify the efficacy of zinc supplementation using biochemical or physiological parameters. The alteration of IgE and the numbers of eosinophils by zinc supplementation should be explored.

Thank you very much for this advice. The sentence referring to the references was unfortunately worded ambiguously. It was written that “studies showed a protective role of zinc supplementation in asthma prevention and symptom improvement” (line 254). However, it is more correctly when phrased “studies showed a protective role of zinc in asthma prevention and symptom improvement” since most of the studies mentioned have not explicitly conducted a supplementation study but found significant correlations of higher zinc levels and less severe symptoms. Hence, the references #78 (now #90), #90 (now #101), #94-98 (now #105-109) in lines xxx are not included in Table 2 but in Table 1. Di Toro et al. published a systematic and meta-analytical review article and did not conduct a study on zinc supplementation. However, they did describe significant effects of zinc supplementation and are therefore mentioned here in the running text. The three studies to which they explicitly refer are also listed in Table 2 and are mentioned in the text by name (Biltagi et al., Ghaffari et al., Rerksuppaphol et al.).

A study by Richter et al. uncovered, that zinc deficiency significantly increases level of bronchopulmonary eosinophilia by 35% (p < 0.05, n = 25). Consistent with that, an increase in eosinophils was found in the perivascular and peribronchial regions of the lung. In contrast, zinc supplementation was found to decrease BAL eosinophils by 34% (p < 0.05, n = 26) (Richter et al.). In line with that Ehlayel et al. showed that hypozincemia, which was observed in 42 (25%, zinc 8.6 ± 1.1 µmoI/L) children, was associated with elevated IgE levels (p = 0.001) (Ehlayel et al.). However, as mentioned in statement #3, studies uncovered that there seem to be only a weak correlation between eosinophilic inflammation and bronchial hyper-responsiveness (Crimi et al., Wislon et al.) and anti-IgE treatment indeed reduces airway eosinophilia but has no effect on bronchial hyper-responsiveness (Djukanović et al.). Since there also seem to be no relationship between the extent of airway remodeling and the degree or duration of any inflammatory parameter (Payne et al.), the numbers of eosinophilia and IgE-expression was not quantified in the text. However, if further concrete findings regarding these connections are uncovered, these should definitely be taken into account in future review articles.

#5. The explanation for Zn-transporters is missing in Figure 1. The authors should note which indicate GPR39 and claudin-3 respectively in Figure 1 (skin epithelia). The explanation for this figure seems to be inadequate. What is Gaq in the figure? The role of metallothionein in the inhibition of caspase-3 activity should be also illustrated in Figure 1 (gastrointestinal epithelia). The authors should cite the reference that zinc supplementation inhibits the activity of caspase-3 by way of the induction of metallothionein.

Thank you so much for your comments. All abbreviations are now explained in the Figure description.

Already, one possible mechanism of caspase inactivation is shown in figure 1 in the gastrointestinal epithelia. Here Zinc administration is demonstrated to directly inhibit caspase activity which is mediated by changing the enzymatic activity of caspase due to direct Zn2+-binding sites present in the enzyme. Hence zinc binding inhibits enzyme activity.

Another mechanism of caspase inactivation is the metallothionein (MT)-mediated modulation. MTs are inducible by various stimuli, like proinflammatory cytokines, glucocorticoids, oxidative stress, and divalent metal cations, such as zinc (Ling et al., Thirumoorthy et al.), which is illustrated in Figure 1 already. Amongst others, MTs are known to modulate cell apoptosis, by having antioxidant properties against reactive oxygen species (ROS) hence protecting cells from DNA damage by scavenging and limiting of ROS (Maret et al., Shimoda et al.). Hence, an indirect negative correlation of MT- and caspase-activity is known which is now illustrated in Figure 1 as well. 

#6. The section for CRS (Page 6 line 251-Page 7 line 297) should have a new subtitle, “Zinc and Chronic Rhinosinusitis”. The correlation between the zinc status and the presence or absence of nasal polyp has not been clearly described in the section. Is the effect of zinc supplementation approximately equivalent between CRSwNP and CRSsNP? Moreover, the figure illustrating the role of zinc in the nasal epithelium is missing.

The new section for Chronic Rhinosinusitis is a very good advice and is added in line271.

Studies exanimating CRS do not necessarily distinguish between CRS with nasal polyps (CRSwNP) or without nasal polyps (CRSsNP) (Unal et al.), nevertheless serum zinc levels were significantly lower in CRS patients compared to healthy controls. One study uncovered decreased zinc levels in nasal mucosa in CRS patients compared to healthy controls, however there were no significant differences in serum zinc levels between CRSsNP, and CRSwNP patients (Murphy et al.). There was a trend however that serum zinc levels in CRSsNP were lower than in CRSwNP patients, but differences did not reach statistical significance. Nevertheless, it is unclear why tissue zinc levels are decreased in nasal mucosa from CRSwNP patients compared to healtyh contols since no significant changes in zinc transporter expressions could be found (Murphy et al, Suzuki et al). In contrast, CRSsNP patients show significantly decreased expression of the zinc transporters ZIP1, ZIP2, ZIP14, ZnT1, and metallothioneins MT1a, and MT2a (Murphy et al., Suzuki et al.).

The nasal epithelium is not shown separately in Figure 2, as CRS and asthma are described in one section and both diseases affect the human respiratory tract. It is represented in Figure 2 as "airway epithelium".

#7. How does GPR39 affect impaired tight junctions by zinc supplementation? The authors should describe a putative mechanism by which GPR39 is involved in the repairment of tight junctions.

Thank you for this good point. The impact of GPRC39 on tight junction stability in relation to zinc status is now added in lines 356-362, highlighted in red).

The zinc-sensing GPR39 is primarily coupled with Gαq subunit of G protein. Upon being agonized, for example by zinc, GPR39 emanates signals mainly via a rise in intracellular calcium concentration through a Gαq-phospholipase C (PLC)-inositol triphosphate (IP3) pathway. Zinc administration provokes GPR39 activation that eventually enhances tight junction assembly. This functions by GPR39-mediated Gαq-phospholipase C (PLC)- Ca(2+)/calmodulin-dependent protein kinase kinase-β (CaMKKβ)-serine/threonine kinase AMP-activated protein kinase (AMPK) (PLC-CaMKKβ-AMPK) pathway activation that enhances the abundance of occludin, thereby improving epithelial integrity (Pongkorpsakol et al., Shao et al.).

#8. The statements on conflict of interest, authors’ contribution, external fund and acknowledgement are missing. These are required items for submitting an article.

 Thank you very much. All mentioned points are included in the article (see lines 505-507 highlighted in red).

Minor

#1. ZA, ZB, ZG and ZO should be spelled out in Table 2, respectively.

Thank you very much. The abbreviations are now explained in the footnote of the table.

#2. What indicates (I) in Table 2? What does Wiegand mean in Table 2?

The explanation of the abbreviations and acronyms used in Table 2 were missing in the footnote. Wiegand et al. is a reference (see below), which was formatted in the wrong way. Everything is now corrected in Table 2.

#3. The abbreviations for Journal can be used in “References”. For instance, Allergy International can be abbreviated as Allergol Int (Reference 116).

Thank you for your kind advice. The reference list was created automatically via the citation program Endnote using the suggested style package of MDPI downloaded of the official MDPI homepage. Therefore, it seems to me as the formatting is desired by the journal and sound not be changed manually.

References

Pongkorpsakol P, Buasakdi C, Chantivas T, Chatsudthipong V, Muanprasat C. An agonist of a zinc-sensing receptor GPR39 enhances tight junction assembly in intestinal epithelial cells via an AMPK-dependent mechanism. Eur J Pharmacol. 2019 Jan 5;842:306-313. doi: 10.1016/j.ejphar.2018.10.038. Epub 2018 Oct 26. PMID: 30459126.

Shao YX, Lei Z, Wolf PG, Gao Y, Guo YM, Zhang BK. Zinc Supplementation, via GPR39, Upregulates PKCζ to Protect Intestinal Barrier Integrity in Caco-2 Cells Challenged by Salmonella enterica Serovar Typhimurium. J Nutr. 2017 Jul;147(7):1282-1289. doi: 10.3945/jn.116.243238. Epub 2017 May 17. PMID: 28515165.

Xue M, Wang Q, Pang B, Zhang X, Zhang Y, Deng X, Zhang Z, Niu W. Association Between Circulating Zinc and Risk for Childhood Asthma and Wheezing: A Meta-analysis on 21 Articles and 2205 Children. Biol Trace Elem Res. 2024 Feb;202(2):442-453. doi: 10.1007/s12011-023-03690-4. Epub 2023 May 5. PMID: 37145255; PMCID: PMC10764583.

Bartemes, K.R.; Kita, H. Roles of innate lymphoid cells (ILCs) in allergic diseases: The 10-year anniversary for ILC2s. J Allergy Clin Immunol 2021, 147, 1531-1547, doi:10.1016/j.jaci.2021.03.015.

Bartemes, K.R.; Kephart, G.M.; Fox, S.J.; Kita, H. Enhanced innate type 2 immune response in peripheral blood from patients with asthma. J Allergy Clin Immunol 2014, 134, 671-678.e674, doi:10.1016/j.jaci.2014.06.024.

Doherty, T.A.; Broide, D.H. Airway innate lymphoid cells in the induction and regulation of allergy. Allergol Int 2019, 68, 9-16, doi:10.1016/j.alit.2018.11.001.

LeSuer WE, Kienzl M, Ochkur SI, Schicho R, Doyle AD, Wright BL, Rank MA, Krupnick AS, Kita H, Jacobsen EA. Eosinophils promote effector functions of lung group 2 innate lymphoid cells in allergic airway inflammation in mice. J Allergy Clin Immunol. 2023 Aug;152(2):469-485.e10. doi: 10.1016/j.jaci.2023.03.023. Epub 2023 Apr 6. PMID: 37028525; PMCID: PMC10503660.

Crimi E, Spanevello A, Neri M, Ind PW, Rossi GA, Brusasco V. Dissociation between airway inflammation and airway hyperresponsiveness in allergic asthma. Am J Respir Crit Care Med. (1998) 157:4–9. 10.1164/ajrccm.157.1.9703002

Wilson NM, James A, Uasuf C, Payne DN, Hablas H, Agrofioti C, et al.. Asthma severity and inflammation markers in children. Pediatr Allergy Immunol. (2001) 12:125–32. 10.1034/j.1399-3038.2001.012003125.x

Djukanović R, Wilson SJ, Kraft M, Jarjour NN, Steel M, Chung KF, et al.. Effects of treatment with anti-immunoglobulin E antibody omalizumab on airway inflammation in allergic asthma. Am J Respir Crit Care Med. (2004) 170:583–93. 10.1164/rccm.200312-1651OC

Berry MA, Hargadon B, Shelley M, Parker D, Shaw DE, Green RH, et al.. Evidence of a role of tumor necrosis factor alpha in refractory asthma. N Engl J Med. (2006) 354:697–708. 10.1056/NEJMoa050580

Payne DN, Rogers AV, Adelroth E, Bandi V, Guntupalli KK, Bush A, et al.. Early thickening of the reticular basement membrane in children with difficult asthma. Am J Respir Crit Care Med. (2003) 167:78–82. 10.1164/rccm.200205-414OC

Ling XB, Wei HW, Wang J, Kong YQ, Wu YY, Guo JL, Li TF, Li JK. Mammalian Metallothionein-2A and Oxidative Stress. Int J Mol Sci. 2016 Sep 6;17(9):1483. doi: 10.3390/ijms17091483. PMID: 27608012; PMCID: PMC5037761.

Thirumoorthy N, Shyam Sunder A, Manisenthil Kumar K, Senthil Kumar M, Ganesh G, Chatterjee M. A review of metallothionein isoforms and their role in pathophysiology. World J Surg Oncol. 2011 May 20;9:54. doi: 10.1186/1477-7819-9-54. PMID: 21599891; PMCID: PMC3114003.

Maret W. Redox biochemistry of mammalian metallothioneins. J Biol Inorg Chem. 2011 Oct;16(7):1079-86. doi: 10.1007/s00775-011-0800-0. Epub 2011 Jun 7. PMID: 21647775.

Shimoda R, Achanzar WE, Qu W, Nagamine T, Takagi H, Mori M, Waalkes MP. Metallothionein is a potential negative regulator of apoptosis. Toxicol Sci. 2003 Jun;73(2):294-300. doi: 10.1093/toxsci/kfg095. Epub 2003 Apr 15. PMID: 12700406.

UnalM,TamerL,PataYS,KilicS,DegirmenciU,AkbasY,etal.Serumlevelsof antioxidant vitamins, copper, zinc and magnesium in children with chronic rhinosinusitis. J Trace Elem Med Biol 2004;18:189e92.

Murphy J, Ramezanpour M, Roscioli E, Psaltis AJ, Wormald PJ, Vreugde S. Mucosal zinc deficiency in chronic rhinosinusitis with nasal polyposis con- tributes to barrier disruption and decreases ZO-1. Allergy 2018;73:2095e7.

Suzuki M, Ramezanpour M, Cooksley C, Lee TJ, Jeong B, Kao S, et al. Zinc- depletion associates with tissue eosinophilia and collagen depletion in chronic rhinosinusitis. Rhinology 2020. https://doi.org/10.4193/ Rhin19.383.

Wiegand C, Hipler UC, Boldt S, Strehle J, Wollina U. Skin-protective effects of a zinc oxide-functionalized textile and its relevance for atopic dermatitis. Clin Cosmet Investig Dermatol. 2013 May 6;6:115-21. doi: 10.2147/CCID.S44865. PMID: 23696710; PMCID: PMC3656624.

Richter M, Bonneau R, Girard MA, Beaulieu C, Larivée P. Zinc status modulates bronchopulmonary eosinophil infiltration in a murine model of allergic inflammation. Chest. 2003 Mar;123(3 Suppl):446S. doi: 10.1378/chest.123.3_suppl.446s. PMID: 12629032.

Ehlayel MS, Bener A. Risk factors of zinc deficiency in children with atopic dermatitis. Eur Ann Allergy Clin Immunol. 2020 Jan;52(1):18-22. doi: 10.23822/EurAnnACI.1764-1489.114. Epub 2019 Oct 8. PMID: 31594297.

NIH; National Institutes of Health. Zinc, Fact Sheet for Health Professionals. Available online: https://ods.od.nih.gov/factsheets/ Zinc-HealthProfessional/ (accessed 08.07.2024).

DGE; Deutsche Gesellschaft für Ernährunf e. V. Referenzwerte für die Nährstoffzufuhr. Available online: https://www.dge.de/ wissenschaft/referenzwerte/?L=0 (accessed 08.07.2024).

World Health Organization (WHO), G., Switzerland. Trace Elements in Human Nutrition and Health. Available online: https://www.who.int/publications/i/item/9241561734 (accessed on 24.05.2024).

EFSA Panel on Dietetic Products, N.; Allergies. Scientific Opinion on Dietary Reference Values for zinc. EFSA Journal 2014, 12, 3844, doi:https://doi.org/10.2903/j.efsa.2014.3844.

Round 2

Reviewer 2 Report

Comments and Suggestions for Authors

The quality of this review article seems to get much better. However, several revisions are still required. Please refer to the comments shown below.

Major

#1. The title of this article does not seem to be impressive to the readers. The keywords that reflect this article have not been appropriately selected.

We thank the reviewer for this note and changed the title to “Zinc deficiency and zinc supplementation in allergic diseases”, which described more clearly its content. The keywords are exhaustive for the content of the review.

Comment: The keywords should be appropriately selected.

#2. The purpose of this study remains uncertain. The authors should clearly describe it in “Introduction”.

 Thank you so much for your notice. A corresponding passage has been inserted in the paragraph “Introduction” lines 49-60 (highlighted in red).

Comment: appropriately responded

#3. The putative mechanisms by which zinc deficiency evokes asthma have not been fully described. The statement on the correlation between zinc deficiency and eosinophilia would be also required in such allergic diseases. Therefore, the authors should revise the Figure 1 (zinc deficiency) to explain the cellular immunity including the role of eosinophils in the pathogenesis of asthma.

Thank you very much for this point. Three mechanisms of the influence of zinc deficiency are already described in this review: Zinc deficiency disturbs the balance between type 1 and type 2 T helpers, resulting in inflammation and eosinophilia. Eosinophilia/Eosinophil cells are now mentioned in lines 150-157, 217, 222, 234/235 and 258/259, 279. New aspects added to the text are highlighted in red. Additionally, zinc deficiency may be associated with the production of IgE, which increases the risk of asthma. Moreover, zinc is of powerful antioxidant activity in the lungs, which may be responsible for the imbalance between oxidation and antioxidant, ultimately reducing antioxidant function and increasing the risk of asthma. (Xue et al.)

Allergic asthma is generally characterized by increased production of type 2 cytokines and accumulation of lymphocytes and eosinophils into the airway. Herein, Group 2 innate lymphoid cells (ILC2s) have been identified as critical upstream mediators of type 2 inflammation and the induced airway eosinophilia in translational models of allergic asthma (Bartemes et al. 2014, Bartemes et al. 2021, Doherty et al.). Although ILC2s are proposed to be upstream mediators of eosinophils, little is known regarding the reciprocal interactions between eosinophils and ILC2s despite their co-occurrence in allergic diseases (LeSuer). This aspect has been added in lines 150-157 (highlighted in red).

Studies uncovered, that there seem to be only a weak correlation at baseline between eosinophilic inflammation and bronchial hyper-responsiveness (Crimi et al., Wislon et al.). The anti-IgE monoclonal omalizumab reduces airway eosinophilia but has no effect on bronchial hyper-responsiveness (Djukanović et al.), whereas the anti-TNF monoclonal etanercept reduces hyper-responsiveness but has no effect on airway inflammation (Berry et al.). Furthermore, there is no relationship between the extent of airway remodeling, specifically reticular basement membrane thickness, and the degree or duration of any inflammatory parameter (Payne et al.). Hence, the role of eosinophilic inflammation and the severity of the disease therefore appears to require further research and details will not be added to the review.

The role of eosinophils in zinc deficiency is now added to Figure 1.

 Comment: The authors respond well to my comment. However, the references written by Crimi et al, Djukanovic et al, Berry et al, and Payne et al. are not found in the text. The authors should reflect the statements described above in the textbook.

#4. The authors should describe the effects of zinc supplementation on asthma more clearly. For instance, the statement that zinc supplementation improved lung parameters such as FVC, FEV1 and FEV1/FVC is found (Page 5, lines233-235). However, the findings are not shown in Table 2. Moreover, the references that zinc supplementation contributed to asthma prevention and symptom improvement are cited [77, 78, 90-98] (Page 5, line 235-236). However, the references #78, #90, #94-#98 are missing in Table 2. The authors should verify the efficacy of zinc supplementation using biochemical or physiological parameters. The alteration of IgE and the numbers of eosinophils by zinc supplementation should be explored.

Thank you very much for this advice. The sentence referring to the references was unfortunately worded ambiguously. It was written that “studies showed a protective role of zinc supplementation in asthma prevention and symptom improvement” (line 254). However, it is more correctly when phrased “studies showed a protective role of zinc in asthma prevention and symptom improvement” since most of the studies mentioned have not explicitly conducted a supplementation study but found significant correlations of higher zinc levels and less severe symptoms. Hence, the references #78 (now #90), #90 (now #101), #94-98 (now #105-109) in lines xxx are not included in Table 2 but in Table 1. Di Toro et al. published a systematic and meta-analytical review article and did not conduct a study on zinc supplementation. However, they did describe significant effects of zinc supplementation and are therefore mentioned here in the running text. The three studies to which they explicitly refer are also listed in Table 2 and are mentioned in the text by name (Biltagi et al., Ghaffari et al., Rerksuppaphol et al.).

A study by Richter et al. uncovered, that zinc deficiency significantly increases level of bronchopulmonary eosinophilia by 35% (p < 0.05, n = 25). Consistent with that, an increase in eosinophils was found in the perivascular and peribronchial regions of the lung. In contrast, zinc supplementation was found to decrease BAL eosinophils by 34% (p < 0.05, n = 26) (Richter et al.). In line with that Ehlayel et al. showed that hypozincemia, which was observed in 42 (25%, zinc 8.6 ± 1.1 µmoI/L) children, was associated with elevated IgE levels (p = 0.001) (Ehlayel et al.). However, as mentioned in statement #3, studies uncovered that there seem to be only a weak correlation between eosinophilic inflammation and bronchial hyper-responsiveness (Crimi et al., Wislon et al.) and anti-IgE treatment indeed reduces airway eosinophilia but has no effect on bronchial hyper-responsiveness (Djukanović et al.). Since there also seem to be no relationship between the extent of airway remodeling and the degree or duration of any inflammatory parameter (Payne et al.), the numbers of eosinophilia and IgE-expression was not quantified in the text. However, if further concrete findings regarding these connections are uncovered, these should definitely be taken into account in future review articles.

 Comment: The reply responds well to the comment. However, unfavorable results by zinc supplementation are not shown in Table 2. The table should be revised again.

In addition, the references written by Richter et al., Ehlayel et al., Djukanovic et al, and Payne et al are not found in the text. The authors should reflect the statements mentioned above in the text.  

#5. The explanation for Zn-transporters is missing in Figure 1. The authors should note which indicate GPR39 and claudin-3 respectively in Figure 1 (skin epithelia). The explanation for this figure seems to be inadequate. What is Gaq in the figure? The role of metallothionein in the inhibition of caspase-3 activity should be also illustrated in Figure 1 (gastrointestinal epithelia). The authors should cite the reference that zinc supplementation inhibits the activity of caspase-3 by way of the induction of metallothionein.

Thank you so much for your comments. All abbreviations are now explained in the Figure description.

Already, one possible mechanism of caspase inactivation is shown in figure 1 in the gastrointestinal epithelia. Here Zinc administration is demonstrated to directly inhibit caspase activity which is mediated by changing the enzymatic activity of caspase due to direct Zn2+-binding sites present in the enzyme. Hence zinc binding inhibits enzyme activity.

Another mechanism of caspase inactivation is the metallothionein (MT)-mediated modulation. MTs are inducible by various stimuli, like proinflammatory cytokines, glucocorticoids, oxidative stress, and divalent metal cations, such as zinc (Ling et al., Thirumoorthy et al.), which is illustrated in Figure 1 already. Amongst others, MTs are known to modulate cell apoptosis, by having antioxidant properties against reactive oxygen species (ROS) hence protecting cells from DNA damage by scavenging and limiting of ROS (Maret et al., Shimoda et al.). Hence, an indirect negative correlation of MT- and caspase-activity is known which is now illustrated in Figure 1 as well. 

 Comment: The reply responds well to the comment. Likewise, the references written by Ling et al, Thirumoorthy et al, and Maret et al. were missing in the text. The statements described above should be inserted in the text. Moreover, there is no explanation for Gaq in the Figure 1.What is Gaq? The authors should explain it in brief.

#6. The section for CRS (Page 6 line 251-Page 7 line 297) should have a new subtitle, “Zinc and Chronic Rhinosinusitis”. The correlation between the zinc status and the presence or absence of nasal polyp has not been clearly described in the section. Is the effect of zinc supplementation approximately equivalent between CRSwNP and CRSsNP? Moreover, the figure illustrating the role of zinc in the nasal epithelium is missing.

The new section for Chronic Rhinosinusitis is a very good advice and is added in line271.

Studies exanimating CRS do not necessarily distinguish between CRS with nasal polyps (CRSwNP) or without nasal polyps (CRSsNP) (Unal et al.), nevertheless serum zinc levels were significantly lower in CRS patients compared to healthy controls. One study uncovered decreased zinc levels in nasal mucosa in CRS patients compared to healthy controls, however there were no significant differences in serum zinc levels between CRSsNP, and CRSwNP patients (Murphy et al.). There was a trend however that serum zinc levels in CRSsNP were lower than in CRSwNP patients, but differences did not reach statistical significance. Nevertheless, it is unclear why tissue zinc levels are decreased in nasal mucosa from CRSwNP patients compared to healtyh contols since no significant changes in zinc transporter expressions could be found (Murphy et al, Suzuki et al). In contrast, CRSsNP patients show significantly decreased expression of the zinc transporters ZIP1, ZIP2, ZIP14, ZnT1, and metallothioneins MT1a, and MT2a (Murphy et al., Suzuki et al.).

The nasal epithelium is not shown separately in Figure 2, as CRS and asthma are described in one section and both diseases affect the human respiratory tract. It is represented in Figure 2 as "airway epithelium".

 Comment: The reply responds appropriately to the comment. The statements described above should be inserted in the text.

#7. How does GPR39 affect impaired tight junctions by zinc supplementation? The authors should describe a putative mechanism by which GPR39 is involved in the repairment of tight junctions.

Thank you for this good point. The impact of GPRC39 on tight junction stability in relation to zinc status is now added in lines 356-362, highlighted in red).

The zinc-sensing GPR39 is primarily coupled with Gαq subunit of G protein. Upon being agonized, for example by zinc, GPR39 emanates signals mainly via a rise in intracellular calcium concentration through a Gαq-phospholipase C (PLC)-inositol triphosphate (IP3) pathway. Zinc administration provokes GPR39 activation that eventually enhances tight junction assembly. This functions by GPR39-mediated Gαq-phospholipase C (PLC)- Ca(2+)/calmodulin-dependent protein kinase kinase-β (CaMKKβ)-serine/threonine kinase AMP-activated protein kinase (AMPK) (PLC-CaMKKβ-AMPK) pathway activation that enhances the abundance of occludin, thereby improving epithelial integrity (Pongkorpsakol et al., Shao et al.).

 Comments: The reply responds very well to the comment. However, the explanation for Gaq is missing.

#8. The statements on conflict of interest, authors’ contribution, external fund and acknowledgement are missing. These are required items for submitting an article.

 Thank you very much. All mentioned points are included in the article (see lines 505-507 highlighted in red).

Comment: appropriately responded.

Minor

#1. ZA, ZB, ZG and ZO should be spelled out in Table 2, respectively.

Thank you very much. The abbreviations are now explained in the footnote of the table.

 Comment: appropriately responded.

#2. What indicates (I) in Table 2? What does Wiegand mean in Table 2?

The explanation of the abbreviations and acronyms used in Table 2 were missing in the footnote. Wiegand et al. is a reference (see below), which was formatted in the wrong way. Everything is now corrected in Table 2.

Comments: The evaluation of zinc supplementation should be performed between zinc-treatment group and non-treatment group. The number of control group is not required in Table 2.

#3. The abbreviations for Journal can be used in “References”. For instance, Allergy International can be abbreviated as Allergol Int (Reference 116).

Thank you for your kind advice. The reference list was created automatically via the citation program Endnote using the suggested style package of MDPI downloaded of the official MDPI homepage. Therefore, it seems to me as the formatting is desired by the journal and sound not be changed manually.

Comment: Several revisions are still required (For instance, references #51, #122, #127, #185). In reference#195, JPEN and J Parenter Enteral Nutr are duplicated.

Comments on the Quality of English Language

Minor correction should be made in severai parts.

Author Response

SECOND REVISION

The quality of this review article seems to get much better. However, several revisions are still required. Please refer to the comments shown below.

Major

#1. The title of this article does not seem to be impressive to the readers. The keywords that reflect this article have not been appropriately selected.

We thank the reviewer for this note and changed the title to “Zinc deficiency and zinc supplementation in allergic diseases”, which described more clearly its content. The keywords are exhaustive for the content of the review.

Comment: The keywords should be appropriately selected.

Thank you very much. The keywords are now corrected.

#2. The purpose of this study remains uncertain. The authors should clearly describe it in “Introduction”.

Thank you so much for your notice. A corresponding passage has been inserted in the paragraph “Introduction” lines 49-60 (highlighted in red).

Comment: appropriately responded

#3. The putative mechanisms by which zinc deficiency evokes asthma have not been fully described. The statement on the correlation between zinc deficiency and eosinophilia would be also required in such allergic diseases. Therefore, the authors should revise the Figure 1 (zinc deficiency) to explain the cellular immunity including the role of eosinophils in the pathogenesis of asthma.

Thank you very much for this point. Three mechanisms of the influence of zinc deficiency are already described in this review: Zinc deficiency disturbs the balance between type 1 and type 2 T helpers, resulting in inflammation and eosinophilia. Eosinophilia/Eosinophil cells are now mentioned in lines 150-157, 217, 222, 234/235 and 258/259, 279. New aspects added to the text are highlighted in red. Additionally, zinc deficiency may be associated with the production of IgE, which increases the risk of asthma. Moreover, zinc is of powerful antioxidant activity in the lungs, which may be responsible for the imbalance between oxidation and antioxidant, ultimately reducing antioxidant function and increasing the risk of asthma. (Xue et al.)

Allergic asthma is generally characterized by increased production of type 2 cytokines and accumulation of lymphocytes and eosinophils into the airway. Herein, Group 2 innate lymphoid cells (ILC2s) have been identified as critical upstream mediators of type 2 inflammation and the induced airway eosinophilia in translational models of allergic asthma (Bartemes et al. 2014, Bartemes et al. 2021, Doherty et al.). Although ILC2s are proposed to be upstream mediators of eosinophils, little is known regarding the reciprocal interactions between eosinophils and ILC2s despite their co-occurrence in allergic diseases (LeSuer). This aspect has been added in lines 150-157 (highlighted in red).

Studies uncovered, that there seem to be only a weak correlation at baseline between eosinophilic inflammation and bronchial hyper-responsiveness (Crimi et al., Wislon et al.). The anti-IgE monoclonal omalizumab reduces airway eosinophilia but has no effect on bronchial hyper-responsiveness (Djukanović et al.), whereas the anti-TNF monoclonal etanercept reduces hyper-responsiveness but has no effect on airway inflammation (Berry et al.). Furthermore, there is no relationship between the extent of airway remodeling, specifically reticular basement membrane thickness, and the degree or duration of any inflammatory parameter (Payne et al.). Hence, the role of eosinophilic inflammation and the severity of the disease therefore appears to require further research and details will not be added to the review.

The role of eosinophils in zinc deficiency is now added to Figure 1.

Comment: The authors respond well to my comment. However, the references written by Crimi et al, Djukanovic et al, Berry et al, and Payne et al. are not found in the text. The authors should reflect the statements described above in the textbook.

Thank you very much. The comment written above, and the references are now included in the text in lines 73-83 (highlighted in blue).

#4. The authors should describe the effects of zinc supplementation on asthma more clearly. For instance, the statement that zinc supplementation improved lung parameters such as FVC, FEV1 and FEV1/FVC is found (Page 5, lines233-235). However, the findings are not shown in Table 2. Moreover, the references that zinc supplementation contributed to asthma prevention and symptom improvement are cited [77, 78, 90-98] (Page 5, line 235-236). However, the references #78, #90, #94-#98 are missing in Table 2. The authors should verify the efficacy of zinc supplementation using biochemical or physiological parameters. The alteration of IgE and the numbers of eosinophils by zinc supplementation should be explored.

Thank you very much for this advice. The sentence referring to the references was unfortunately worded ambiguously. It was written that “studies showed a protective role of zinc supplementation in asthma prevention and symptom improvement” (line 254). However, it is more correctly when phrased “studies showed a protective role of zinc in asthma prevention and symptom improvement” since most of the studies mentioned have not explicitly conducted a supplementation study but found significant correlations of higher zinc levels and less severe symptoms. Hence, the references #78 (now #90), #90 (now #101), #94-98 (now #105-109) in lines xxx are not included in Table 2 but in Table 1. Di Toro et al. published a systematic and meta-analytical review article and did not conduct a study on zinc supplementation. However, they did describe significant effects of zinc supplementation and are therefore mentioned here in the running text. The three studies to which they explicitly refer are also listed in Table 2 and are mentioned in the text by name (Biltagi et al., Ghaffari et al., Rerksuppaphol et al.).

A study by Richter et al. uncovered, that zinc deficiency significantly increases level of bronchopulmonary eosinophilia by 35% (p < 0.05, n = 25). Consistent with that, an increase in eosinophils was found in the perivascular and peribronchial regions of the lung. In contrast, zinc supplementation was found to decrease BAL eosinophils by 34% (p < 0.05, n = 26) (Richter et al.). In line with that Ehlayel et al. showed that hypozincemia, which was observed in 42 (25%, zinc 8.6 ± 1.1 µmoI/L) children, was associated with elevated IgE levels (p = 0.001) (Ehlayel et al.). However, as mentioned in statement #3, studies uncovered that there seem to be only a weak correlation between eosinophilic inflammation and bronchial hyper-responsiveness (Crimi et al., Wislon et al.) and anti-IgE treatment indeed reduces airway eosinophilia but has no effect on bronchial hyper-responsiveness (Djukanović et al.). Since there also seem to be no relationship between the extent of airway remodeling and the degree or duration of any inflammatory parameter (Payne et al.), the numbers of eosinophilia and IgE-expression was not quantified in the text. However, if further concrete findings regarding these connections are uncovered, these should definitely be taken into account in future review articles.

Comment: The reply responds well to the comment. However, unfavorable results by zinc supplementation are not shown in Table 2. The table should be revised again.

In addition, the references written by Richter et al., Ehlayel et al., Djukanovic et al, and Payne et al are not found in the text. The authors should reflect the statements mentioned above in the text.

Thank you very much. The comment written above, and the references are now included in the text in lines 265-273 (highlighted in blue).

So far, unfavorable results of zinc supplementation regarding allergic diseases are not known, hence they are not included in Tab. 2. Nevertheless, zinc deficiency and appearance or exacerbation of allergic diseases is known, which is shown in Tab. 2.

#5. The explanation for Zn-transporters is missing in Figure 1. The authors should note which indicate GPR39 and claudin-3 respectively in Figure 1 (skin epithelia). The explanation for this figure seems to be inadequate. What is Gaq in the figure? The role of metallothionein in the inhibition of caspase-3 activity should be also illustrated in Figure 1 (gastrointestinal epithelia). The authors should cite the reference that zinc supplementation inhibits the activity of caspase-3 by way of the induction of metallothionein.

Thank you so much for your comments. All abbreviations are now explained in the Figure description.

Already, one possible mechanism of caspase inactivation is shown in figure 1 in the gastrointestinal epithelia. Here Zinc administration is demonstrated to directly inhibit caspase activity which is mediated by changing the enzymatic activity of caspase due to direct Zn2+-binding sites present in the enzyme. Hence zinc binding inhibits enzyme activity.

Another mechanism of caspase inactivation is the metallothionein (MT)-mediated modulation. MTs are inducible by various stimuli, like proinflammatory cytokines, glucocorticoids, oxidative stress, and divalent metal cations, such as zinc (Ling et al., Thirumoorthy et al.), which is illustrated in Figure 1 already. Amongst others, MTs are known to modulate cell apoptosis, by having antioxidant properties against reactive oxygen species (ROS) hence protecting cells from DNA damage by scavenging and limiting of ROS (Maret et al., Shimoda et al.). Hence, an indirect negative correlation of MT- and caspase-activity is known which is now illustrated in Figure 1 as well.

Comment: The reply responds well to the comment. Likewise, the references written by Ling et al, Thirumoorthy et al, and Maret et al. were missing in the text. The statements described above should be inserted in the text. Moreover, there is no explanation for Gaq in the Figure 1.What is Gaq? The authors should explain it in brief.

Thank you for your comment. The comment written above, and the references are now included in the text in lines 373-378 (highlighted in blue). Gαq, the G protein alpha q subunit, is now included in line 196 (highlighted in blue).

#6. The section for CRS (Page 6 line 251-Page 7 line 297) should have a new subtitle, “Zinc and Chronic Rhinosinusitis”. The correlation between the zinc status and the presence or absence of nasal polyp has not been clearly described in the section. Is the effect of zinc supplementation approximately equivalent between CRSwNP and CRSsNP? Moreover, the figure illustrating the role of zinc in the nasal epithelium is missing.

The new section for Chronic Rhinosinusitis is a very good advice and is added in line271.

Studies exanimating CRS do not necessarily distinguish between CRS with nasal polyps (CRSwNP) or without nasal polyps (CRSsNP) (Unal et al.), nevertheless serum zinc levels were significantly lower in CRS patients compared to healthy controls. One study uncovered decreased zinc levels in nasal mucosa in CRS patients compared to healthy controls, however there were no significant differences in serum zinc levels between CRSsNP, and CRSwNP patients (Murphy et al.). There was a trend however that serum zinc levels in CRSsNP were lower than in CRSwNP patients, but differences did not reach statistical significance. Nevertheless, it is unclear why tissue zinc levels are decreased in nasal mucosa from CRSwNP patients compared to healtyh contols since no significant changes in zinc transporter expressions could be found (Murphy et al, Suzuki et al). In contrast, CRSsNP patients show significantly decreased expression of the zinc transporters ZIP1, ZIP2, ZIP14, ZnT1, and metallothioneins MT1a, and MT2a (Murphy et al., Suzuki et al.).

The nasal epithelium is not shown separately in Figure 2, as CRS and asthma are described in one section and both diseases affect the human respiratory tract. It is represented in Figure 2 as "airway epithelium".

Comment: The reply responds appropriately to the comment. The statements described above should be inserted in the text.

Thank you for your advice. The mentioned statements are now inserted in the text in lines 295/296, 305-306, 311-316 (highlighted in blue).

#7. How does GPR39 affect impaired tight junctions by zinc supplementation? The authors should describe a putative mechanism by which GPR39 is involved in the repairment of tight junctions.

Thank you for this good point. The impact of GPRC39 on tight junction stability in relation to zinc status is now added in lines 356-362, highlighted in red).

The zinc-sensing GPR39 is primarily coupled with Gαq subunit of G protein. Upon being agonized, for example by zinc, GPR39 emanates signals mainly via a rise in intracellular calcium concentration through a Gαq-phospholipase C (PLC)-inositol triphosphate (IP3) pathway. Zinc administration provokes GPR39 activation that eventually enhances tight junction assembly. This functions by GPR39-mediated Gαq-phospholipase C (PLC)- Ca(2+)/calmodulin-dependent protein kinase kinase-β (CaMKKβ)-serine/threonine kinase AMP-activated protein kinase (AMPK) (PLC-CaMKKβ-AMPK) pathway activation that enhances the abundance of occludin, thereby improving epithelial integrity (Pongkorpsakol et al., Shao et al.).

Comments: The reply responds very well to the comment. However, the explanation for Gaq is missing.

The explanation for Gαq is now included (see line 196 highlighted in blue).

#8. The statements on conflict of interest, authors’ contribution, external fund and acknowledgement are missing. These are required items for submitting an article.

Thank you very much. All mentioned points are included in the article (see lines 505-507 highlighted in red).

Comment: appropriately responded.

Minor

#1. ZA, ZB, ZG and ZO should be spelled out in Table 2, respectively.

Thank you very much. The abbreviations are now explained in the footnote of the table.

Comment: appropriately responded.

#2. What indicates (I) in Table 2? What does Wiegand mean in Table 2?

The explanation of the abbreviations and acronyms used in Table 2 were missing in the footnote. Wiegand et al. is a reference (see below), which was formatted in the wrong way. Everything is now corrected in Table 2.

Comments: The evaluation of zinc supplementation should be performed between zinc-treatment group and non-treatment group. The number of control group is not required in Table 2.

Thank you very much for your comment. In Table 2, some studies are listed without having a standard control group that receives e.g. placebo treatment. Instead, patients are either treated first with zinc and then not treated, or vice versa. Thus, this group itself serves as both a control and an intervention group during the course of the study. The explanation is now inserted in the Table description (see line 545 highlighted in blue).

#3. The abbreviations for Journal can be used in “References”. For instance, Allergy International can be abbreviated as Allergol Int (Reference 116).

Thank you for your kind advice. The reference list was created automatically via the citation program Endnote using the suggested style package of MDPI downloaded of the official MDPI homepage. Therefore, it seems to me as the formatting is desired by the journal and sound not be changed manually.

Comment: Several revisions are still required (For instance, references #51, #122, #127, #185). In reference#195, JPEN and J Parenter Enteral Nutr are duplicated.

Thank you very much for this comment. All references are now inserted in the abbreviated form (highlighted in blue):

#13 Pediatric Allergy and Immunology à Pediatr Allergy Immunol
#56 The American Journal of Clinical Nutrition à Am J Clin Nutrition,

#91 Egyptian Journal of Chest Diseases and Tuberculosis à Egypt J Chest Dis Tuberc

#99 Journal of Krishna Institute of Medical Sciences University à JKIMSU

#101 Pediatric Pulmonology à Pediatr Pulmonol

#111 International Journal of Current Research and Review àIJCRR

#114 The Iraqi Postgraduate Medical  à Journal Iraq Med. J 2

#121 The Medical Journal of Mashhad University of Medical Sciences à mjms

#122 The Turkish Journal of Biochemistry  àTJB 

#124 Sudan Journal of Medical Sciences à SJMS

#128 Frontiers in Pharmacology à Front Pharmacol

#133 Allergology international à Allergol Int

#179 Immunology & Cell Biology à ICB

#188 Iraqi Journal of Medical Sciences à Iraqi J Med Sci

#196 Iranian Journal of Dermatology à Iran J Dermatol

#197 Benha Journal of Applied Sciences à BJAS

#202 Journal of Clinical & Experimental Dermatology Research à Clin Exp Dermatol

#204 J Parenter Enteral Nutr à JPEN

Round 3

Reviewer 2 Report

Comments and Suggestions for Authors

Reviewer’s comment

The authors responded very well to the all major comments, However, in Table 2, “C” indicates health control. However, it is not true. “Healthy control” should be replaced with untreated group or placebo group. In addition, Do the authors know that the reference#108 was retracted? The numbers of treated group and untreated group in the reference #110 are 144 and 140, respectively. The numbers should be corrected in the table.

Author Response

We thank the reviewer for the points mentioned and changed the manuscript in all points:

Comment: The authors responded very well to the all major comments, However, in Table 2, “C” indicates health control. However, it is not true. “Healthy control” should be replaced with untreated group or placebo group.

Response: "C" was changed to "untreated control"

Comment: In addition, Do the authors know that the reference#108 was retracted?

Response: We thank the reviewer for this hint. It is now indicated that reference 108 was retracted in 2012.

Comment: The numbers of treated group and untreated group in the reference #110 are 144 and 140, respectively. The numbers should be corrected in the table.

Response: Many thanks to the reviewer for indicating our mistake. We corrected the numbers in the table.